# DASHCO: Data-Aware SAT Heuristics Combinations Optimization via Large Language Models

## Abstract

The performance of Conflict-Driven Clause Learning solvers hinges on internal heuristics, yet the heterogeneity of SAT problems makes a single, universally optimal configuration unattainable. While prior automated methods can find specialized configurations for specific problem families, this dataset-specific approach lacks generalizability and requires costly re-optimization for new problem types. We introduce DASHCO, a framework that addresses this challenge by learning a generalizable mapping from instance features to tailored heuristic ensembles, enabling a train-once, adapt-broadly model. Our framework uses a Large Language Model, guided by systematically defined Problem Archetypes, to generate a diverse portfolio of specialized heuristic ensembles and subsequently learns an adaptive selection mechanism to form the final mapping. Experiments show that DASHCO achieves superior performance and, most notably, demonstrates robust out-of-domain generalization where non-adaptive methods show limitations. Our work establishes a more scalable and practical path toward automated algorithm design for complex, configurable systems.

## 1 Introduction

The Boolean Satisfiability (SAT) problem is a cornerstone of computational complexity theory Cook (2023) and a problem of immense practical importance Crawford & Baker (1994). Its applications are vast, with modern solvers enabling crucial advances in diverse fields such as formal verification Prasad et al. (2005), planning Rintanen (2012), and program analysis Harris et al. (2010). While SAT is NP-complete, powerful solvers based on the Conflict-Driven Clause Learning (CDCL) paradigm Audemard & Simon (2009) can often solve massive industrial instances with remarkable efficiency. However, this success is not uniform. The performance of these solvers is critically sensitive to the internal algorithmic heuristics that guide their search. Developing effective heuristics has consequently been a central focus of SAT research for decades Audemard & Simon (2012); Liang et al. (2018), traditionally relying on a long and costly process of manual design.

The challenge of designing effective heuristics has led to the development of hyper-heuristics Burke et al. (2013), which automate the process of selecting or generating algorithms. With the advent of LLMs Achiam et al. (2023); Comanici et al. (2025), this field has seen a surge of innovation. Frameworks like FunSearch Romera-Paredes et al. (2024), Evolution of Heuristics (EoH) Liu et al. (2024), and ReEvo Ye et al. (2024) have demonstrated that LLMs can generate novel and effective heuristics for various combinatorial optimization (CO) problems by treating algorithm design as a program search task within an evolutionary framework van Stein & Bäck (2024). While the recent AutoSAT framework Sun et al. (2024) has made important progress by applying this paradigm to SAT solvers, a significant limitation underlies these pioneering works: they are inherently dataset-specific. Their methodology is tailored to a particular training distribution, yielding a single, static heuristic configuration optimized for that specific class of problems. Consequently, the resulting solver lacks generalizability, and the expensive search process must be repeated for each new problem family.

This limitation is particularly severe because the SAT problem space is enormously heterogeneous, serving as a universal language for problems from diverse domains like Minesweeper Kaye (2000), cryptographic analysis Soos et al. (2009), and logistics planning Rintanen (2012). Moreover, the

performance of a complex system like a CDCL solver is determined by the intricate and often non-intuitive **interplay** of its multiple, interacting heuristics for tasks such as restarts Audemard & Simon (2012); Liang et al. (2018) and phase selection. The synergistic or conflicting effects of these combinations are notoriously difficult to predict, a challenge that even human experts struggle to navigate. This implies that simply selecting the best-performing heuristic for each component in isolation is unlikely to yield a globally optimal solver. It is therefore crucial to holistically explore the vast combinatorial space of heuristic configurations, rather than merely optimizing individual components. The goal of such exploration should not be to find a single "best" configuration, as the optimal choice is deeply instance-dependent given the problem space's heterogeneity Ansótegui et al. (2009). Rather, the challenge is to develop a system that can adapt its strategy based on the problem at hand.

To address these fundamental challenges of generalizability and adaptation, we propose **DASHCO (Data-Aware SAT Heuristics Combinations Optimization)**. Instead of seeking a single, universally optimal solver, DASHCO introduces a new paradigm: its objective is to learn a generalizable mapping from instance characteristics to tailored heuristic configurations. The core of our framework is its data-aware nature, which enables a train-once, adapt-broadly model. DASHCO first leverages an LLM, guided by high-level Problem Archetypes, to automatically generate a diverse portfolio of specialized heuristic ensembles, each tailored to different problem structures. Subsequently, it learns an adaptive selection mechanism that partitions the instance space based on performance, creating a map to dynamically choose the best-suited ensemble for any new SAT instance. This approach transforms the problem from finding one optimal point to learning a function over the entire problem space.

We summarize our main contributions as follows:

- We introduce DASHCO, a novel framework that shifts the paradigm from dataset-specific optimization to generalizable, data-aware algorithm design, directly addressing the critical limitation of prior work.
- We propose a methodology where Problem Archetypes guide an LLM to generate a diverse portfolio of specialized heuristic ensembles, enabling an automated exploration of the complex interactions between different heuristic components.
- Through extensive experiments, we demonstrate that DASHCO significantly outperforms baselines. Crucially, it exhibits superior **out-of-domain generalization** compared to methods that learn a single, non-adaptive configuration, validating the effectiveness and robustness of our approach.

## 2  PRELIMINARIES & RELATED WORK

### 2.1  SAT AND CDCL-BASED SOLVERS

Let $V = \{x_1, \ldots, x_n\}$ be a finite set of Boolean variables. The corresponding set of literals is defined as $L = V \cup \{\neg v \mid v \in V\}$. A *clause* $C$ is a finite subset of $L$, representing the disjunction of its literals, with the constraint that for any variable $v \in V$, both $v$ and $\neg v$ cannot be simultaneously present in $C$. A formula $F$ in Conjunctive Normal Form (CNF) is a set of clauses, $\{C_1, \ldots, C_m\}$, representing their conjunction. A *truth assignment* (or interpretation) is a function $\tau : V \to \{\top, \bot\}$ that maps each variable to a truth value. The satisfaction of a formula under $\tau$, denoted by $\tau \models F$, is defined hierarchically:

1. A literal $l \in L$ is satisfied by $\tau$ ($\tau \models l$) if $l = v$ and $\tau(v) = \top$, or if $l = \neg v$ and $\tau(v) = \bot$.
2. A clause $C$ is satisfied by $\tau$ ($\tau \models C$) if there exists at least one literal $l \in C$ such that $\tau \models l$.
3. A formula $F$ is satisfied by $\tau$ ($\tau \models F$) if for all clauses $C_i \in F$, $\tau \models C_i$.

The Boolean Satisfiability (SAT) problem is the computational task of determining whether a given CNF formula $F$ is satisfiable, i.e., whether there exists any truth assignment $\tau$ such that $\tau \models F$.

Modern SAT solvers are predominantly based on the Conflict-Driven Clause Learning (CDCL) framework. A CDCL-based solver iteratively builds a partial assignment by making decisions and

applying Boolean Constraint Propagation (BCP). When a conflict arises, the solver analyzes the cause, learns a new clause to prevent the same conflict from recurring, and backtracks. This process is profoundly influenced by a complex interplay of internal heuristics. These strategies guide key aspects of the search, including which variable to decide on next (branching), when to abandon an unpromising search path and restart, and how to diversify the search by adjusting variable phases. The specific combination and implementation of these heuristics are what differentiate modern solvers and are the primary determinants of their performance.

## 2.2 LLM-BASED HEURISTIC GENERATION

The emergence of LLMs has opened a new frontier for automated algorithm design. By leveraging their powerful code generation and reasoning capabilities, researchers have started to automate the discovery of heuristics. A prominent approach is to frame heuristic generation as a program search problem within an evolutionary framework. FunSearch established this paradigm by evolving programs to find new mathematical discoveries Romera-Paredes et al. (2024). This was extended to CO problems by frameworks like EoH Liu et al. (2024) and ReEvo Ye et al. (2024), which use evolutionary algorithms to prompt an LLM to iteratively refine and improve heuristic code for problems like TSP and online bin packing. ReEvo notably introduced a reflective step, where the LLM provides textual feedback to guide the evolutionary search, emulating a verbal gradient Ye et al. (2024). MEoH Yao et al. (2025) models automated heuristic design as a multi-objective optimization problem, using a dominance-dissimilarity mechanism with an LLM to generate a set of heuristics that balance performance and efficiency.

AutoSAT Sun et al. (2024) was the first to apply this paradigm to the intricate environment of SAT solvers. Recognizing that generating a competitive solver from scratch is infeasible due to code complexity, AutoSAT proposed a modular framework where an LLM optimizes specific, predefined heuristic functions within an existing solver. It successfully demonstrated that an LLM could enhance a baseline CDCL solver to achieve competitive performance.

While these methods are powerful, they generally produce a single, universally applied heuristic or configuration, overlooking the instance-specific nature of algorithm performance. An emerging paradigm in automated algorithm design seeks to overcome this limitation by partitioning a problem class into subclasses based on instance features. This allows for the creation of specialized heuristics tailored to the unique characteristics of each subclass. Our work, DASHCO, applies this principle of data-awareness to the multi-heuristic, complex environment of CDCL solvers. In doing so, we bridge the gap between the universal optimization of frameworks like AutoSAT and a more granular, data-centric approach to algorithm design.

## 2.3 ALGORITHM SELECTION AND PORTFOLIO SOLVERS

A major paradigm for tackling instance heterogeneity is portfolio-based algorithm selection, pioneered by the influential SatZilla framework Xu et al. (2008). SatZilla leverages a portfolio of diverse, human-designed solvers and uses machine learning models to predict the best-performing one for a given instance based on its features. DASHCO inherits this data-driven philosophy but introduces a fundamental novelty: rather than selecting from a portfolio of pre-existing solvers, it first uses an LLM to *automatically generate* a new portfolio of fine-grained heuristic ensembles. DASHCO is thus not only an algorithm selector but also an automated portfolio generator, a key distinction that significantly expands the space of possible solver configurations.

## 3 METHODOLOGY

To address the challenge of dataset-specific optimization and the lack of generalizability in prior work, we propose DASHCO. The core of our methodology is a paradigm shift: instead of repeatedly executing an expensive search for a single, specialized solver for each new problem family, our goal is to construct a single, robust, and adaptive framework that generalizes across them.

Specifically, DASHCO's objective is not to find one *best* heuristic ensemble, but to learn a rich *mapping* from the instance feature space to the space of high-performance heuristic configurations. This is achieved by first creating a diverse portfolio of specialized heuristic ensembles and then learn-

Figure 1: An overview of the DASHCO framework, illustrating the three primary stages: (1) Data-Aware Heuristic Evolution, (2) Instance Space Partitioning, and (3) Adaptive Heuristic Selection.

ing an intelligent selection mechanism. Once constructed, this framework can be deployed on new, unseen SAT instances from diverse problem families, dynamically selecting a suitable configuration without the need for re-optimization. This "train-once, adapt-broadly" approach is designed to be more practical and efficient for real-world applications. The framework operates in three main stages: (1) Data-Aware Heuristic Evolution, to build the rich portfolio of heuristic components; (2) Instance Space Partitioning, to learn the mapping between instance types and optimal ensembles; and (3) Adaptive Heuristic Selection, to apply this mapping to new instances.

## 3.1 HEURISTIC MODULES IN CDCL-BASED SOLVERS

Following the modular design of AutoSAT, we focus on optimizing a set of critical, independent heuristic functions within a CDCL-based solver. For this work, we target three key heuristics:

1. **Restart Policy (`restart`):** This module implements a crucial strategy to prevent the solver from becoming trapped in unproductive regions of the search space. A restart policy determines when to abandon the current search path and backtrack to the top decision level. While the current partial assignment is discarded, all learned clauses are retained, allowing the solver to begin a new search attempt with more information. Modern policies are often dynamic, adapting their restart frequency to the search progress to effectively combat heavy-tailed runtime distributions Luby et al. (1993); Audemard & Simon (2012).

2. **Phase Selection (`rephase`):** This heuristic acts as a diversification mechanism by managing the default polarity (true or false) assigned to variables. During branching, the solver often uses a variable's last assigned value as a default choice. Over time, these saved phases can lead to search stagnation. The rephase function is called periodically to alter these saved phases, for instance by resetting them or flipping them. This forces the solver to explore different branches first, effectively diversifying the search and pushing it into new areas of the solution space Jeroslow & Wang (1990).

3. **Variable Bumping (`bump_var`):** This module is a core component of adaptive branching strategies. When the solver encounters a conflict, it analyzes the implication graph to identify the variables responsible for the contradiction. The bump_var function is then invoked to increase a numerical score, often called an 'activity score', associated with each of these variables. By elevating the scores of conflict-prone variables, this mechanism dynamically influences the main branching heuristic to prioritize them in future decisions.

This effectively focuses the search on the most constrained and active parts of the problem space Biere et al. (2009); Liang et al. (2016).

Our goal is to find not just one optimal implementation for each of these, but a diverse set of effective implementations that can be combined into powerful ensembles.

## 3.2 FEATURE EXTRACTION FOR SAT INSTANCES

A critical precursor to any data-aware method is the ability to characterize instances with a quantitative feature vector. We define a 37-dimensional feature vector $v(j)$ that maps a given SAT instance $j$ to a vector in $\mathbb{R}^{37}$, designed to capture a comprehensive set of its structural and statistical properties.

Our feature set is constructed by augmenting an established feature template with new global descriptors. The foundation consists of the complete 33-feature set adapted from the well-established algorithm selection framework SATzilla, excluding only the most computationally expensive features that require pre-solving. To further enhance the descriptive power, we then introduced 4 additional lightweight, high-level features that capture global properties such as overall polarity bias and constraint density. This approach combines a proven, powerful feature set with novel global metrics, balancing expressive capability with the efficiency required for a dynamic, per-instance selection model. A detailed breakdown of all 37 features is provided in the Appendix.

## 3.3 DATA-AWARE HEURISTIC EVOLUTION

To build our library of heuristics, we introduce a guided search process centered around a set of predefined **Problem Archetypes**. The purpose of these archetypes is to guide the LLM to search for heuristics along high-level, human-understandable directions. This ensures that the resulting heuristics are differentiated, each specialized for a particular type of problem structure, which is crucial for building a powerful and diverse portfolio for later combination.

**Defining Problem Archetypes and Search Environments.** A key design choice is to define these archetypes manually (e.g., 'highly-constrained problems' or 'instances with heterogeneous clause structures'), rather than using automated clustering at this stage. This is because our goal is to create effective, semantic prompts to guide the LLM. These interpretable concepts serve as more potent guidance for the LLM's creative process than the abstract centroids produced by a black-box clustering algorithm.

These archetypes serve a crucial dual role in creating specialized search environments. For each textual archetype $d_i$, we first leverage the LLM to identify a relevant subset of features from our 38-dimensional space that best characterize it. The corresponding data subset $I_i \subseteq I_{train}$ is then curated by splitting the training set at the 50% threshold based on the values of these LLM-selected features. This process creates a targeted training and evaluation environment for that specific archetype. Simultaneously, the textual description of the archetype is injected as a direct hint into the LLM's prompt to guide the creative code generation, for example: `Please note that we are focusing on highly-constrained problems.`

**Guided Evolutionary Search.** Let $\mathcal{D} = \{(d_i, I_i)\}_{i=1}^p$ be the set of established pairs, where each $d_i$ is an optimization direction and $I_i$ is its corresponding data subset. For each pair $(d_i, I_i) \in \mathcal{D}$, we conduct an independent evolutionary search for each heuristic module. During this search, the textual hint from $d_i$ directs the LLM to generate heuristics specialized for that direction, with their performance evaluated exclusively on the dedicated data subset $I_i$. After this guided search, the total set of generated heuristic ensembles, formed by the Cartesian product $\mathcal{H} = \mathcal{L}_{restart} \times \mathcal{L}_{rephase} \times \mathcal{L}_{bump\_var}$, undergoes a pruning step where low-performing combinations are filtered out, resulting in a smaller, high-quality portfolio $\mathcal{H}'$.

## 3.4 INSTANCE SPACE PARTITIONING VIA PERFORMANCE-BASED CLUSTERING

The Cartesian product of generated heuristics can yield a large number of candidate ensembles, posing a computational challenge for the partitioning stage. To manage this, we constrain the number of components per heuristic type (e.g., $k \leq 3$) and perform an aggressive pruning step. All generated

ensembles are evaluated on a benchmark subset, and only a fixed number of the top-performing candidates are retained in the final portfolio, $\mathcal{H}'$.

Given this pruned portfolio of high-quality ensembles $\mathcal{H}'$, the next stage is to understand which ensemble is best for which kind of instance. To achieve this, we partition our training set of instances, $I_{train}$, based on performance.

First, we evaluate every heuristic ensemble $h_i \in \mathcal{H}$ on every instance $j \in I_{train}$. Let $p(h_i, j)$ be the performance metric (e.g., PAR-2 score) of ensemble $h_i$ on instance $j$.

Next, for each instance $j$, we identify its optimal ensemble, $h^*(j)$, from our library:

$$h^*(j) = \arg \min_{h_i \in \mathcal{H}} p(h_i, j) \tag{1}$$

This allows us to partition the instance set $I_{train}$ into disjoint clusters, where each cluster $C_i$ is associated with a single best-performing ensemble $h_i$:

$$C_i = \{j \in I_{train} \mid h^*(j) = h_i\} \tag{2}$$

This process creates a direct mapping from a region in the instance feature space (represented by the instances in $C_i$) to an optimal solver configuration $h_i$. For each resulting cluster $C_i$, we compute its feature space centroid, $\bar{v}_i$, by averaging the feature vectors of all its member instances.

### 3.5 Adaptive Heuristic Selection for Instances

With the partitioned instance space and associated optimal ensembles, we can now perform adaptive selection for any new, unseen test instance $j_{new}$. We first extract its feature vector $v(j_{new})$. We then calculate the distance (using normalized Euclidean distance) between $v(j_{new})$ and each cluster centroid $\bar{v}_i$. The heuristic ensemble $h_k$ associated with the closest centroid $\bar{v}_k$ is selected as the most suitable configuration for solving $j_{new}$. This allows the solver to dynamically adapt its strategy based on the data characteristics of the problem at hand.

## 4 Experiments

To evaluate the effectiveness of DASHCO, we conduct a series of experiments designed to assess its performance against baseline and state-of-the-art SAT solvers.

### 4.1 Experimental Setup

**Environment and Parameters.** All solvers are implemented in C++ and compiled with g++ 12.3.0. The LLM interaction and the evolutionary framework are managed in Python. Experiments were conducted on servers equipped with AMD Ryzen 9 5950X 16-core processors and 128GB of RAM. For all heuristic generation tasks, we utilized the GPT-4o model with a temperature of 0.8 to encourage diverse outputs. Considering the computational expense of solving SAT problems and the scale of our benchmarks, the timeout for solving any single SAT instance was set to 1000 seconds.

**Backbone Solver.** To ensure a methodologically consistent and comparable basis, we follow the precedent set by AutoSAT Sun et al. (2024) in selecting our backbone solver. Our framework is built upon *EasySAT*, a lightweight and modular CDCL solver. As established in prior work, this choice provides a clean and capable baseline that is well-suited for modification by LLMs, striking a practical balance between solver functionality and the token-context limitations inherent in current LLM-based code generation.

**Datasets.** Our experimental design is structured to rigorously evaluate the generalization capabilities of DaSAThco. We construct a single, heterogeneous training set by combining 24 unclassified instances from the SAT Competitions of 2022 and 2023 with 10 instances from each of the *Coins-Grid*, *LangFord*, and *PRP* benchmarks. Our evaluation then proceeds in two distinct settings:

- **In-Domain Generalization:** We test performance on held-out instances from the *Coins-Grid*, *LangFord*, and *PRP* families, which were partially represented in the training set.

Table 1: Main performance comparison across In-Domain and Out-of-Domain datasets. For each solver, we report both the PAR-2 score (lower is better) and the number of solved instances (Solved, higher is better). The timeout is 1000s. Best results in each category are in bold.

| Dataset | # Inst. | EasySAT | | MiniSat | | AutoSAT | | DASHCO | | Kissat | |
|---|---|---|---|---|---|---|---|---|---|---|---|
| | | PAR-2 | #S | PAR-2 | #S | PAR-2 | #S | PAR-2 | #S | PAR-2 | #S |
| *In-Domain Benchmarks* | | | | | | | | | | | |
| CoinsGrid | 52 | 1757.5 | 7 | 1985.1 | 4 | 1711.1 | 9 | **1399.5** | **16** | 1404.2 | **16** |
| LangFord | 64 | 1915.9 | 4 | 2000.0 | 0 | 1850.2 | 8 | 1750.9 | 12 | **1610.1** | **14** |
| PRP | 144 | 1970.4 | 3 | 1935.8 | 9 | 1843.8 | 15 | 1739.6 | 20 | **1570.0** | **38** |
| *Out-of-Domain Benchmarks* | | | | | | | | | | | |
| CNP | 50 | 594.9 | 38 | 1150.2 | 22 | 614.4 | 39 | 550.7 | 41 | **270.7** | **44** |
| Zamkeller | 48 | 830.5 | 30 | 1950.6 | 5 | 764.8 | 32 | 611.4 | 35 | **24.8** | **48** |
| KnightTour | 22 | 1733.4 | 3 | 1900.7 | 2 | 1684.6 | 4 | **1369.0** | **7** | 1638.5 | 4 |

- **Out-of-Domain Generalization:** To evaluate performance on entirely novel problem structures, we use test sets from the *CNP (Chromatic Number of the Plane)*, *Zamkeller*, and *KnightTour* families, which were completely excluded from the training process.

**Baselines.** We compare the performance of DASHCO against a set of representative baselines:

- **EasySAT**: The lightweight, modular CDCL solver that serves as the direct backbone for our modifications. Its performance represents the starting point before any LLM-based optimization.

- **AutoSAT** Sun et al. (2024): A state-of-the-art framework representing the prior paradigm of dataset-specific optimization. To ensure a fair comparison of generalization capabilities, we adapt its methodology to our experimental setting. Instead of running its search process on each individual problem family, we run AutoSAT on the same single, heterogeneous training set used by DASHCO. This evaluates its ability to find a single "average-best" configuration for a diverse set of problems. Consequently, the performance reported here is not directly comparable to its original publication, where it was optimized on specialized datasets, and may be lower.

- **MiniSat** Sorensson & Een (2005): A classic and highly-optimized CDCL solver, which serves as a robust and widely-recognized traditional baseline.

- **Kissat** Biere et al. (2024): A state-of-the-art, highly-engineered CDCL solver that represents the pinnacle of modern manual heuristic design and frequently wins SAT competitions. It serves as a top-tier benchmark for performance. We deploy the 4.0.0 version of kissat which submitted to the SAT Competition 2024 and won 3 gold medals.

**Metrics.** We evaluate solver performance using two standard metrics in the SAT competition: the number of solved instances within the timeout, and the Penalized Average Runtime with a factor of 2 (PAR-2) score. The PAR-2 score is the average runtime across a set of instances, but with a heavy penalty for any instance that is not solved within the 1000s timeout. Specifically, unsolved instances are assigned a runtime of twice the timeout (2000s).

## 4.2 Performance Comparison

We evaluate the performance of DASHCO against the baselines on both in-domain and out-of-domain datasets, with detailed results presented in Table 1. The findings clearly demonstrate the effectiveness of our data-aware, portfolio-based approach and offer critical insights when compared against different design paradigms.

Overall, DASHCO consistently and significantly outperforms the EasySAT, MiniSat, and AutoSAT baselines across both evaluation settings. The comparison with AutoSAT is particularly illuminating, as it confirms that for a diverse problem set, our adaptive selection from a specialized portfolio is more effective than relying on a single, average-best configuration produced by prior paradigms.

The comparison with Kissat highlights the core strengths of our framework. As expected, Kissat, a top-tier solver, shows exceptional performance on many benchmarks, particularly on well-known families like Zamkeller. However, the goal of our work is not to surpass a manually-honed solver on every problem, but to demonstrate a more generalizable and automated design paradigm. The results strongly support this goal. Most notably, on the out-of-domain KnightTour dataset, DASHCO decisively outperforms all other solvers, including Kissat. This result is a powerful validation of our central thesis: for novel problem structures, a static set of highly-tuned heuristics can be suboptimal, whereas DASHCO's adaptive mechanism can dynamically select a more suitable configuration from its generated portfolio. Furthermore, on the in-domain CoinsGrid benchmark, DASHCO achieves performance on par with Kissat, demonstrating that our automated framework can generate and select configurations that are competitive with the state-of-the-art.

These results lead to a clear conclusion: while highly-engineered solvers like Kissat represent the peak of performance for specific problem distributions, our automated, data-aware framework presents a robust and promising path towards building solvers that can generalize more effectively across a wide and unpredictable landscape of SAT instances.

## 4.3 ABLATION STUDY

To understand the contribution of the key components of our framework, we conduct an ablation study on two representative datasets: CoinsGrid for in-domain generalization, and Zamkeller for out-of-domain (OOD) generalization. We compare our full DASHCO model against several ablated variants:

- **w/o Data-Aware Generation:** The portfolio is generated without the guidance of Problem Archetypes. The adaptive selection is then applied to this non-specialized portfolio.

- **Random Selection:** For each instance, we randomly select an ensemble from the fully generated, specialized portfolio.

- **Single Best Selection:** We select the single ensemble that performs best on average across the entire training set and apply it to all instances, simulating a non-adaptive paradigm.

- **Oracle:** This represents the theoretical performance upper bound of our portfolio, where for each test instance, we assume an oracle perfectly selects the best-performing ensemble from our generated portfolio.

The results are presented in Table 2. The full DASHCO model achieves the best performance among all practical variants. The importance of our data-aware generation is evident when comparing the full model to the 'w/o Data-Aware Generation' variant, which shows a clear performance drop.

The analysis of the selection mechanism reveals a crucial insight. The *Single Best Selection* approach performs significantly worse than the adaptive DASHCO, particularly on the OOD dataset Zamkeller, highlighting the failure of a non-adaptive strategy to generalize. Interestingly, the 'Oracle' results demonstrate that there is still considerable room for improvement in the selection mechanism. The gap between DASHCO and the Oracle suggests that while our nearest-centroid selector is effective, more sophisticated selection models could unlock even greater performance from the generated portfolio, representing a promising avenue for future work. Nonetheless, both the data-aware portfolio generation and the dynamic selection mechanism are shown to be critical to DASHCO's robust performance.

## 4.4 ANALYSIS OF PORTFOLIO SCALE AND DIVERSITY

The size of the heuristic portfolio is a critical hyperparameter in our framework. We focused our sensitivity analysis on the two most influential modules: restart and bump_var, fixing the rephase component to its default implementation. We evaluated the performance of DASHCO on the CoinsGrid dataset while varying the portfolio cardinalities for these two heuristics ($k_{restart}$ and $k_{bump\_var}$), with k ranging from 0 to 3.

Figure 2 presents a heatmap of the results, which reveals several key insights. First, a significant performance improvement is observed when moving from a single heuristic ($k = 1$ for either dimension) to combinations of multiple diverse heuristics ($k \geq 2$). For instance, the performance of the

Table 2: Ablation study on a representative In-Domain (CoinsGrid) and Out-of-Domain (Zamkeller) dataset. We report the PAR-2 score (lower is better) and the number of solved instances (#S, higher is better).

|  | CoinsGrid | | Zamkeller | |
|---|---|---|---|---|
| **Model Variant** | PAR-2 | #S | PAR-2 | #S |
| **DASHCO** | **1399.5** | **16** | **611.4** | **35** |
| w/o Data-Aware | 1671.0 | 10 | 722.4 | 33 |
| Random | 1740.2 | 8 | 780.5 | 29 |
| Single Best | 1503.9 | 12 | 764.8 | 32 |
| Oracle | 1157.8 | 19 | 467.6 | 41 |

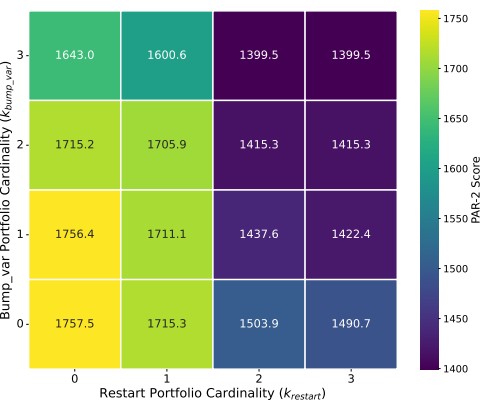

Figure 2: Sensitivity analysis of DASHCO's performance on the CoinsGrid dataset. The heatmap shows the average PAR-2 score as a function of the portfolio cardinalities for the restart ($k_{restart}$) and bump_var ($k_{bump\_var}$) heuristics. The k=0 point corresponds to the EasySAT baseline.

portfolio with ($k_{restart} = 2, k_{bump\_var} = 2$) is substantially better than portfolios with only one varied component, such as ($k_{restart} = 2, k_{bump\_var} = 1$). This demonstrates the powerful synergistic effects that arise from combining a diverse set of specialized heuristics, validating the core premise of our portfolio-based approach. Second, the heatmap shows that simply increasing the number of heuristics does not guarantee monotonic improvement, highlighting the complex interplay between them. The optimal performance is achieved with the portfolio of size ($k_{restart} = 2, k_{bump\_var} = 3$) or ($k_{restart} = 3, k_{bump\_var} = 3$). This illustrates that while a richer portfolio is generally beneficial, the quality and compatibility of the added heuristics are crucial. The results confirm that a diverse, multi-component portfolio is essential for achieving top performance.

### 4.5 OVERHEAD ANALYSIS

We analyze the computational overhead of DASHCO in terms of its one-time offline costs and per-instance online costs. The offline cost involves generating and compiling the heuristic portfolio, a process made manageable by the lightweight EasySAT backbone. The per-instance online overhead for solving a new instance is negligible, consisting of two fast operations: feature extraction and adaptive selection. The calculation of the feature vector is computationally inexpensive and can be pre-computed for known benchmarks. Subsequently, our adaptive selection via a nearest-centroid search is extremely efficient, consistently taking less than a second. This minimal online cost makes DASHCO highly practical. Future work could explore the trade-off between this efficient selector and more sophisticated models, such as LLMs.

### 5 CONCLUSION

In this work, we introduced DASHCO, a novel framework that addresses the critical generalizability limitations in automated SAT solver design by shifting from dataset-specific optimization to a scalable "train-once, adapt-broadly" paradigm. Our methodology leverages a Large Language Model, guided by Problem Archetypes, to generate a diverse portfolio of specialized heuristic ensembles and then learns an adaptive mechanism to select the best configuration for new instances. Experiments confirm that this approach not only improves performance but, more importantly, exhibits robust out-of-domain generalization. This validates that learning a mapping from instance features to a generated portfolio of solvers is a more effective and practical paradigm for the automated design of complex, configurable systems like SAT solvers. Future work could focus on automating the discovery of these archetypes and extending this paradigm to other domains.

REPRODUCIBILITY STATEMENT

To ensure the reproducibility of our work, we commit to making our source code publicly available upon acceptance of this paper. All SAT instances used for training and evaluation are from publicly available SAT Competition benchmarks and previous work. We will provide detailed lists and instructions for obtaining them in our repository. More details of our implementation can be found in the Appendix.

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

## A    DETAILED SAT INSTANCE FEATURES

Our 37-dimensional feature vector is constructed from a combination of 33 features adapted from the SATzilla template and four new lightweight additions. The complete list is detailed below.

- **Problem Size Features**
    - num_variables: The number of variables, denoted as $v$.
    - num_clauses: The number of clauses, denoted as $c$.
    - var_clause_ratio: The ratio of clauses to variables ($c/v$).
- **Variable-Clause Graph Features**
    - Variable Nodes Degree (Frequency): avg_var_frequency, var_degree_cv, var_degree_min, max_var_frequency, var_frequency_entropy.

- Clause Nodes Degree (Length): `avg_clause_length`, `clause_degree_cv`, `min_clause_length`, `max_clause_length`, `clause_length_entropy`.

- **Variable Graph Features**

  - Nodes Degree Statistics: `var_graph_degree_mean`, `var_graph_degree_cv`, `var_graph_degree_min`, `var_graph_degree_max`.

- **Balance and Polarity Features (Distributional)**

  - Per-Clause Literal Ratio: `clause_pos_neg_ratio_mean`, `clause_pos_neg_ratio_cv`, `clause_pos_neg_ratio_entropy`.
  - Per-Variable Occurrence Ratio: `var_pos_neg_ratio_mean`, `var_pos_neg_ratio_cv`, `var_pos_neg_ratio_min`, `var_pos_neg_ratio_max`, `var_pos_neg_ratio_entropy`.
  - Clause Type Fraction: `binary_clause_fraction`, `ternary_clause_fraction`.

- **Proximity to Horn Formula Features**

  - `horn_clause_fraction`: The fraction of clauses that are Horn clauses.
  - Horn Variable Occurrences: `horn_var_occurrence_mean`, `horn_var_occurrence_cv`, `horn_var_occurrence_min`, `horn_var_occurrence_max`, `horn_var_occurrence_entropy`.

- **Additional High-Level Features (Global)**

  - `positive_literal_ratio`: The global ratio of positive literals to total literals.
  - `balanced_var_ratio`: The proportion of variables with an equal number of positive and negative occurrences.
  - `polarity_bias`: A measure of the overall tendency towards positive or negative literals.
  - `constraint_density`: A measure of problem constraint level.

# B  ALGORITHM OF DASHCO

---

**Algorithm 1** DASHCO Framework Overview

---

1: **Input:** Training instance set $I_{train}$, test instance $j_{new}$.
2: **Output:** Solved result for $j_{new}$.
      ▷ Stage 1: Data-Aware Heuristic Evolution
3: Define a set of Problem Archetypes $D = \{d_1, ..., d_p\}$.
4: **for** each archetype $d_i \in D$ **do**
5:     Create data subset $I_i \subseteq I_{train}$ by filtering.
6: **end for**
7: $\mathcal{H} \leftarrow \mathcal{L}_{restart} \times \mathcal{L}_{rephase} \times \mathcal{L}_{bump\_var}$
8: $\mathcal{H}' \leftarrow \text{Prune}(\mathcal{H}, I_{train})$ by removing ensembles with performance below a predefined threshold.
      ▷ Stage 2: Offline Instance Space Partitioning
9: **for** each instance $j \in I_{train}$ **do**
10:     $h^*(j) \leftarrow \arg\min_{h_k \in \mathcal{H}'} \text{performance}(h_k, j)$.
11: **end for**
12: **for** each unique $h_k$ that is optimal for some instance **do**
13:     $C_k \leftarrow \{j \in I_{train} | h^*(j) = h_k\}$.
14:     $\overline{v}_k \leftarrow \frac{1}{|C_k|} \sum_{j \in C_k} v(j)$.
15: **end for**
      ▷ Stage 3: Online Adaptive Selection
16: $v_{new} \leftarrow v(j_{new})$.
17: $k \leftarrow \arg\min_i \text{distance}(v_{new}, \overline{v}_i)$.
18: $h_{selected} \leftarrow h_k$.
19: Solve $j_{new}$ using the solver configured with $h_{selected}$.

---

## C   DATASET CHARACTERISTICS

Our experimental design aims to evaluate the generalization capability of DASHCO. We construct a single, heterogeneous **training set** composed of instances from multiple sources, including unclassified problems from the SAT Competitions of 2022 and 2023, as well as the SCPC family. Our evaluation is then divided into two settings:

- **In-Domain Generalization:** We test on unseen instances from families that were partially represented in the training set. These benchmarks include *CoinsGrid*, which originates from a puzzle about arranging coins on a grid under specific constraints; *LangFord*, a combinatorial challenge of arranging paired numbers such that each pair of number $k$ is separated by exactly $k$ other items; and *PRP (Profitable Robust Production)*, which models an industrial task of finding a robust production plan under uncertainty. We select 10 instances from those datasets to compose our training set.

- **Out-of-Domain (OOD) Generalization:** To evaluate true generalization capabilities, we use entire problem families that were completely held out during the heuristic evolution process. These OOD test sets include *CNP (Chromatic Number of the Plane)*, a classic graph coloring problem; *Zamkeller*, a complex permutation problem concerning subsequences; and *KnightTour*, which seeks a path for a knight to visit every square on a chessboard exactly once.

Table 3: Statistical characteristics of the evaluation datasets.

| Dataset | # Inst. | Variables (Mean ± Std) | Clauses (Mean ± Std) |
|---|---|---|---|
| *In-Domain Benchmarks* | | | |
| CoinsGrid | 52 | 530807 ± 513663 | 3825868 ± 3701594 |
| LangFord | 64 | 312492 ± 213284 | 2734786 ± 1972624 |
| PRP | 144 | 499206 ± 324889 | 3337426 ± 2175367 |
| *Out-of-Domain Benchmarks* | | | |
| CNP | 50 | 9890 ± 11139 | 86724 ± 77379 |
| Zamkeller | 48 | 21435 ± 19119 | 265218 ± 283330 |
| KnightTour | 22 | 135288 ± 191062 | 5742107 ±9872215 |

## D   DEFINITION OF PAR-2 METRIC

The PAR-2 score is calculated as:

$$\text{PAR-2} = \frac{1}{N} \sum_{i=1}^{N} t_i'$$

where $N$ is the number of instances, and $t_i'$ for instance $i$ is its actual runtime if solved, or 2000s if unsolved. A lower PAR-2 score is better, as it indicates a solver is both fast and robust. Crucially, the PAR-2 score also serves as the primary fitness metric that guides the LLM-driven evolutionary search for better heuristics.

## E   EXAMPLE PROMPT TEMPLATE

Our data-aware guidance mechanism is designed to be agnostic to the specific underlying LLM-based heuristic search algorithm. It can be integrated as a modular hint into various existing frameworks, such as AutoSAT or ReEvo. The following provides an example of how our data-aware hint can be incorporated into an *Advisor* prompt, using a structure similar to that of AutoSAT. The key addition is the highlighted text, which provides the LLM with the specific Problem Archetype it should optimize for.

---

**Example Advisor Prompt Template**

You are a SAT solver researcher trying to write the {{ task }} to help SAT solver escape from local optimum. Your goal is to write a {{ task }} for the SAT solver that will help it restart the search and escape from local optimum, after reading and understanding the <key_code> of SAT solver below. **Please note, the SAT problem instances we are targeting have the following characteristics: {{ problem_archetype_description }}.**
Your answer must follow the following JSON format:

```
{
    "description":
        "Provide a ...",
    "modification_direction":
        ["some possible directions..."]
}
```

**Tips:**

1. You must traverse all possible positions of <key_code> if you want to modify the {{ task }}.

2. You need to give us some advice to modify the {{ task }}. e.g. some potential directions to change the heuristics.

3. Notice that, you can only change {{ task }}.

4. {{ other_tips }}

**key_code of SAT solver is:**

```
"""
{{ origin_key_code }}
"""
```

Take a deep breath and think it step by step. Then respond strictly in JSON format!

---

## F    HYPERPARAMETER SETTINGS

Table 4 lists the main hyperparameters used across our experiments to ensure reproducibility.

Table 4: Main hyperparameters used in our experiments.

| Hyperparameter | Value |
|---|---|
| *General Experimental Settings* | |
| Solver Timeout | 1000s |
| Random Seed | 42 |
| *LLM and Evolutionary Search* | |
| LLM Model | GPT-4o |
| Temperature | 0.8 |
| Generations | 3 |
| Population Size | 2 |
| *DASHCO Framework Settings* | |
| Number of Problem Archetypes | 3 |
| Heuristics per Type ($k$) | 3 |
| Pruned Portfolio Size | 9 |

## G    USAGE OF LLMS

We take advantage of LLMs to improve the writing of this paper.

