# OpenReview forum: "DASHCO: Data-Aware SAT Heuristics Combinations Optimization via Large Language Models"
_ICLR.cc/2026/Conference — ICLR 2026 Conference Withdrawn Submission_

### Official Review · Reviewer_ouR3 · 2025-10-21

**Soundness:** 3
**Presentation:** 3
**Contribution:** 2
**Rating:** 6
**Confidence:** 4

**Summary:**

This paper studies SAT problem. As claimed by the authors, previous work lack generalizability. In this work, they propose a framework that adaptively generates heuristic sequences for problem instances. The authors conduct experiments to evaluate the effectiveness of their approach. They claim that their proposed method addresses the key generality limitation in the design of automated SAT solvers.

**Strengths:**

1. The paper is generally well-written, with clear explanations.

2. This paper focus on framework studying, which is a foundational work in SAT solving.

3. Clear and explicative figures.

**Weaknesses:**

1. Specifically, what types of Problem Archetypes are included? Table 4 shows only three types of Problem Archetypes. Can they cover various SAT problems?

2. Compared with Kissat, the advantages of this method are not significant, and considering the use of lightweight solvers as the backbone, it is worth considering whether the framework can introduce performance improvements over complex solvers.

**Questions:**

Please reply to my comments listed in ‘Weaknesses’.

**Details Of Ethics Concerns:**

N/A.

---

### Official Review · Reviewer_5cbM · 2025-10-31

**Soundness:** 3
**Presentation:** 4
**Contribution:** 3
**Rating:** 6
**Confidence:** 3

**Summary:**

This paper introduces DASHCO (Data-Aware SAT Heuristics Combinations
Optimization), a framework that leverages a Large Language Model (LLM)
to automatically generate specialized heuristic ensembles for CDCL-based SAT
solvers. Instead of optimizing a single solver configuration, DASHCO constructs
a diverse heuristic portfolio guided by textual Problem Archetypes, evaluates
these on instance subsets, and learns a mapping from instance features to the

best-performing ensemble. The framework thus enables adaptive solver con-
figuration for new, unseen SAT instances. Experiments on several benchmark

families show that DASHCO outperforms baselines such as AutoSAT and Min-
iSat and even achieves competitive or superior results to Kissat on some out-of-
domain datasets.

**Strengths:**

• Clear and well-motivated attempt to bridge data-aware algorithm selection with LLM-based heuristic generation.
• Strong empirical performance, including some out-of-domain generalization improvements.
• Sound modular design that aligns with established SAT optimization paradigms.
• Elegant integration of textual archetypes as semantically interpretable guidance for heuristic diversification.

**Weaknesses:**

• The necessity of the LLM is not convincingly demonstrated; all ablations still rely on GPT-4o, so the advantage over non-LLM heuristic search remains unclear.
• The benchmark families used (CoinsGrid, LangFord, PRP, CNP, Zamkeller, KnightTour) are relatively niche; it is unclear how diverse or challenging they are compared to industrial SAT benchmarks.

**Questions:**

Can the authors provide stronger evidence that the heuristics generated
by GPT-4o are not simply recombinations of known strategies (e.g., from Kissat or other open-source solvers) that the model may have seen during pretraining? For example, have the authors compared the generated heuristics’ code or logic patterns against existing solver implementations?

Would the framework still perform competitively if the heuristic portfolio H were constructed from a fixed set of top-performing or manually curated heuristics, rather than LLM-generated ones? In other words, can the authors quantify how much of the performance gain truly depends on the LLM’s generative capabilities?

---

### Official Review · Reviewer_9SCZ · 2025-10-31

**Soundness:** 1
**Presentation:** 2
**Contribution:** 1
**Rating:** 2
**Confidence:** 4

**Summary:**

The paper studies using LLMs and text descriptions to generate code for pre-defined heuristic methodologies (restarts, phase, bumping) in a SAT solver  (easysat). During training, a set of these heuristics are generated which can then be mapped into clusters based on distance in a fixed set of instance features. At runtime, closest cluster's heuristics are used for solving.

Experiments compare against SAT solvers (traditional minisat, modern kissat), baseline easysat vs easysat-dashco, and autsat. The comparison is split to in-domain and out-of-domain. There is an ablation on some design choices.

**Strengths:**

- The idea of using LLMs to generate/evolve heuristics has become an emerging trend and the paper focuses on that in SAT domain, which is nice.
- The idea of textual descriptors specific to problem characteristics is a good blend of human intuition - domain knowledge with generative LLM.
- The approach might have potential in improving automated SAT solving (but not necessarily "best" existing SAT solvers, as also noted in the paper)
- I like the focus on "practically". Several design choices (e.g., lightweight extractors) are in the direction of fast/pragmatic solving.

**Weaknesses:**

Overall, I like reading this paper and here are a number areas for improvement.

- It is not clear to me how the Problem Archetypes are decided? This is a crucial part of the overall approach, and the presentation is not clear/precise. How  many are there, 3? How are they decided? Is this manually generated? Or, is this LLM generated (if so, how?).

- Likewise, it is not clear to me how does the Evolution of these heuristics work? Is this using a population-based approach whereby LLM is used to generate individuals in the population. How is it initialized, evolved, mutated, cross-overed, for how long, etc.? IF, this is not the case, how does the LLM heuristic generation works? Is it just a single-shot prompt?

- The idea of learning instance characteristics to solver/heuristic configuration is not new at all. The current write-up risks coming across as the first "data-aware" approach, which is not true. I would recommend tuning down this part. Instance specific algorithm configuration has been exceedingly successful in many domains, including SAT. An immediate reference, that's unfortunately missing and should be referred is ISAC Instance-Specific Algorithm Configuration (ECAI'10). The paper rightly mentions SATzilla, and there are few others that is required for context; these include Hydra (AAAI'10) which builds portfolios for SAT, Non-Model-Based Algorithm Portfolios for SAT (SAT'11), Algorithm selection and scheduling (CP'12). Especially ISAC is relevant in its clustering based configuration selection --very similar to the approach here. Similar examples exists beyond SAT, too, e.g., Automatic MILP solver configuration by learning problem similarities (Annals of OR'24). What's emerging is offloading the heuristic generation to LLMs, as opposed to algorithm tuners (which btw should also be cited, like GG - Gender-Based Genetic Algorithm for the Automatic Configuration of Algorithms (CP'09) or ParamILS (AAAI'07) or SMAC (Hutter et. al.). The paper is written as-if these work do not exist. Please consider (major) revision in the future versions, while pointing out the new exciting directions with generative LLMs started by FunSearch.

- Algorithmic question: why does your approach creates heuristics independently and then requires an "aggresive" pruning in the Cartesian combination which is found to be computationally expense. Have you considered generating a complete configuration with all three heuristics at once in combination? This might be a good ablation to consider. As-is, this seems like an arbitrary choice, and requires extra filtering, compute etc.

- There is a comment about the choice of the SAT solver (easySAT) and token-context limitation. How are the two related to each other?

- Training Dataset: How does the 24 unclassified instances from competitions are selected? This seems arbitrary, no?

- Test Set: how do you select CNP, Zamkeller, KTour for testing? why these domains? What's connection between these datasets and your problem archetypes? Why are the problem archetypes not related to problem domain descriptions?

- Without further details the selection of Training and Test sets seems very ad-hoc and might perceived as fit for method purposes.

- The method is referred to as DaSAThco vs. DASHco sometimes? Is this a typo or some other variants are considered?

- It seems unfortunate that AutoSAT is not tuned per problem for a fair/better comparison.

- There is a section on Overhead Analysis but it does not express anything about what that overhead is, it just "states" that there is overhead? Please revise this section.

- Btw, I really liked your pragmatic SAT feature extractor. If possible to release this, would be excellent to share with the community, given its  lightweight nature. Side note you refer to it as 37-dimensional and sometimes as 38-dimensional, pls fix

- Experimental Comparisons are lacking. We do not have good control to understand/evaluate the added value of the LLM-based heuristic generation brings. The comparison with a traditional (minisat) and modern (kissat) solver is nice. I agree with the authors that we should not expect this to beat the best solver out there (it would be amazing but I would not hold that against the approach if it's behind). one might ask why not directly evolve Kissat in the first place? But I understand that from a "methodology" perspective. Then, what matters is the comparison/improvement achieved on top of the base solver that this work is built upon, which takes us to EasySAT vs. DashCo. Here we see improvements which is in the right direction. BUT, how do we know the improvements are due to the LLM generated heuristic selection. There is no good control in between. The fact that the performance of a single shot EasySAT will be improved with a portfolio of new easysat versions is totally expected. How much of that we can attribute to LLMs and this approach, I don't know.. Have you tried running another portfolio next to this? Say a static ensemble easysat configs OR a bundle of sat solvers against DashCo. That would be a way to go. The comparison with AutoSAT helps, but it's only averaged tuned on all problems and not by domain which is very unfortunate. At the very least, I would recommend comparing EasySAT vs. EasySAT-Tuned vs. DashCo, where EasySAT-tuned is a tuned/portfolio version using GGA, SMAC, ParamILS, Bayesian Opt, Optune, HyperOpt (or any other tuner you like) within the same runtime overhead that training DashCo takes (which we are not given btw..)

- The most interesting part would be the analysis of the methods. What does Kissat do for restarts, phasing, bumping? What new methods LLMs design? How many problem archetypes are there? How is that related to initial problem domains and their selection? How many clusters we have, what are their centroids refer to etc, how do they related to problem archetypes or benchmark domains? I am very surprised that this is not discussed/shared at all.

- A revision version of this paper would be excellent fit to SAT conference

- It's not clear to me what's meant by "50% threshold(s) were applied". Is this for normalized features or not? How stable is this for randomization/re-runs or problem domain. What features are picked by LLMs (this would be super interesting to share but there is a lot of details that the paper is not telling the reader)

- Btw, any reason to use Par-2 as opposed to standard Par-10?

- final pruned portfolio size appears to be 9. That seems arbitrary, too. How is that decided?

**Questions:**

See the weakness

---

### Official Review · Reviewer_AEvD · 2025-11-01

**Soundness:** 3
**Presentation:** 2
**Contribution:** 2
**Rating:** 4
**Confidence:** 4

**Summary:**

The paper proposes a framework for generating SAT solver heuristics
automatically with LLMs offline to use them in portfolios. The authors describe
their approach and evaluate it empirically, comparing to a number of other SAT
solvers.

**Strengths:**

The paper proposes an interesting approach to generating SAT solvers that relies
on a large amount of offline work to, when actually solving problems, be able to
rely on a large number of diverse heuristics. The empirical results show that
this seems to work quite well in practice.

**Weaknesses:**

I did not understand the problem archetypes. The authors say that they are based
on "human-understandable" descriptions, but do not explain how they were
obtained and how many there are. Further, the heuristics generated are then
clustered by problem type, making it even less clear why this was not also done
for the problem instances. It is also unclear whether the human-designed
descriptions are more meaningful with respect to performance and generating
heuristics than a clustering based on instance features -- as instance features
are clearly related to the performance of solvers on the instance, I would
conjecture that this is not the case. Finally, it was unclear to me whether this
actually has any advantage for prompting the LLM. More explanation of this would
be appreciated.

As far as I am able to tell, the authors never say how the generated portfolios
are actually run -- all solvers in parallel? If so, this needs to be taken into
account, as more resources are provided to the authors' approach. It is
certainly not surprising that more solver runs produce better results -- could
the same be achieved with different EasySAT etc runs (e.g. different parameter
settings)?

The proposed approach is conceptually similar to ISAC [1] -- this should be
discussed.

This paper might be better submitted to the SAT conference.

[1] Serdar Kadioglu et al., "ISAC – Instance-Specific Algorithm Configuration," 19th European Conference on Artificial Intelligence (Amsterdam, The Netherlands, The Netherlands), IOS Press, 2010, 751–56, http://portal.acm.org/citation.cfm?id=1860967.1861114.

**Questions:**

See above.

---

### Note · Authors · 2025-12-05

**Comment:**

To the Area Chairs and Reviewers:

We have decided to withdraw our submission. We sincerely thank the reviewers for their time and detailed feedback.

In particular, we appreciate the constructive criticism regarding the Problem Archetype methodology. We acknowledge that the concerns raised by multiple reviewers about the definition and usage of archetypes are valid and point to a need for more rigorous analysis.

We believe that properly addressing these fundamental issues requires a substantial revision that extends beyond the scope of the current rebuttal phase. We plan to use these valuable insights to improve the work significantly for a future submission.

Thank you again for your efforts.

**Withdrawal Confirmation:**

I have read and agree with the venue's withdrawal policy on behalf of myself and my co-authors.